# Resilience of three-dimensional sinusoidal networks in liver tissue

**Jens Karschau**[1], **André Scholich**[2], **Jonathan Wise**[1,3], **Hernán Morales-Navarrete**[4], **Yannis Kalaidzidis**[4], **Marino Zerial**[4,5], **Benjamin M. Friedrich**[1,5]*

**1** cfaed, TU Dresden, Dresden, Germany, **2** Max Planck Institute for the Physics of Complex Systems, Dresden, Germany, **3** Univ. Grenoble Alpes, CNRS, LPMMC, Grenoble, France, **4** Max Planck Institute of Molecular Cell Biology and Genetics, Dresden, Germany, **5** Cluster of Excellence 'Physics of Life', TU Dresden, Dresden, Germany

* benjamin.m.friedrich@tu-dresden.de

**Data Availability Statement:** All relevant data are within the manuscript and its Supporting Information files.

## Abstract

Can three-dimensional, microvasculature networks still ensure blood supply if individual links fail? We address this question in the sinusoidal network, a plexus-like microvasculature network, which transports nutrient-rich blood to every hepatocyte in liver tissue, by building on recent advances in high-resolution imaging and digital reconstruction of adult mice liver tissue. We find that the topology of the three-dimensional sinusoidal network reflects its two design requirements of a space-filling network that connects all hepatocytes, while using shortest transport routes: sinusoidal networks are sub-graphs of the Delaunay graph of their set of branching points, and also contain the corresponding minimum spanning tree, both to good approximation. To overcome the spatial limitations of experimental samples and generate arbitrarily-sized networks, we developed a network generation algorithm that reproduces the statistical features of 0.3-mm-sized samples of sinusoidal networks, using multi-objective optimization for node degree and edge length distribution. Nematic order in these simulated networks implies anisotropic transport properties, characterized by an empirical linear relation between a nematic order parameter and the anisotropy of the permeability tensor. Under the assumption that all sinusoid tubes have a constant and equal flow resistance, we predict that the distribution of currents in the network is very inhomogeneous, with a small number of edges carrying a substantial part of the flow—a feature known for hierarchical networks, but unexpected for plexus-like networks. We quantify network resilience in terms of a permeability-at-risk, i.e., permeability as function of the fraction of removed edges. We find that sinusoidal networks are resilient to random removal of edges, but vulnerable to the removal of high-current edges. Our findings suggest the existence of a mechanism counteracting flow inhomogeneity to balance metabolic load on the liver.

## Author summary

The liver is the largest metabolic organ of the human body and pivotal for blood detoxification and the uptake of many medically relevant drugs. Inside liver tissue, a dense meshwork of blood micro-vessels, the sinusoidal network, contacts every hepatocyte cell. The

**Funding:** JK, MZ and BMF were supported by the Deutsche Forschungsgemeinschaft (DFG, German Research Foundation) under Germany´s Excellence Strategy – EXC-1056 – 194636624– Cluster of Excellence cfaed of TU Dresden. MZ and BMF were supported by the Deutsche Forschungsgemeinschaft (DFG, German Research Foundation) under Germany´s Excellence Strategy – EXC-2068 – 390729961– Cluster of Excellence Physics of Life of TU Dresden. This work was further financially supported by the German Federal Ministry of Education and Research (BMBF) (LiSyM: grant #031L0038 to MZ), European Research Council (ERC) (grant #695646 to MZ) and the Max Planck Society (MPG) (YK and MZ). JW acknowledges funding by the DAAD RISE program. The funders had no role in study design, data collection and analysis, decision to publish, or preparation of the manuscript.

**Competing interests:** The authors have declared that no competing interests exist.

architecture of these networks, and especially how their function responds to local damage is not well understood. Previous theoretical work addressed network resilience in simple two-dimensional networks such as vein networks of leaves but not in three-dimensional networks, partly due to the considerable difficulties of imaging three-dimensional tissues. Here, we build on unprecedented advances in imaging mouse liver tissue. By simulating arbitrarily-sized networks that faithfully reproduce the statistical features of spatially restricted experimental samples, we computationally characterize the relation between a weak alignment of the network along the flow direction and enhanced transport along this direction. A simple transport model predicts an inhomogeneous distribution of flow in the sinusoidal network. Concomitantly, these networks are resilient against the random removal of edges, but vulnerable against removal of 'highway' edges that carried a high flow in the unperturbed state. We speculate that yet unknown adaptive mechanisms balance the distribution of flows and thereby increase the resilience of this physiologically important network.

## Introduction

Leaf venation [1], fungal mycelium [2, 3], but also animal trails networks [4], river deltas [5], and even force networks in granular materials [6], each represent natural transport networks formed by self-organization. Inside our body, the micro-vasculature forms a plexus-like, three-dimensional network of small capillaries that deliver nutrients to every cell in a tissue and removes waste products [7].

Mathematically, transport networks are spatial networks, i.e., graphs embedded in space. It has been proposed that the presence of cycles in these graphs provides redundancy, and thereby resilience against the failure of individual links [8]. Yet, to the best of our knowledge, this concept has never been tested in three-dimensional, micro-vasculature networks. Past research addressed resilience properties almost exclusively in two-dimensional biological transport networks. It was shown that self-organization by local feedback rules can generate hierarchical networks resembling those of leaf networks, which optimize flow resistance upon removal of a single link [1]. For example, work on two-dimensional networks addressed the balance between the cost of network formation and network resilience to random failure [9], the cost of repair after perturbations [10], or adaptation to fluctuations in load [11].

The restriction to two-dimensional networks in past research was largely a consequence of the considerable difficulties in imaging three-dimensional networks. Yet, topology suggests fundamental differences between two-dimensional and three-dimensional spatial networks, as it imposes constraints on the distribution of cycles in the network [12]. Here, we take advantage of recent technological advances in high-resolution imaging of adult mouse liver tissue [13, 14], allowing us to study statistical geometry and resilience of three-dimensional sinusoidal networks.

The liver is the largest metabolic organ in the human body. It is responsible for storage of metabolites, secretion of digestion enzymes and detoxification of blood [7]. The liver is organized into millimeter-sized basic functional units termed lobules, see Fig 1A. Each lobule comprises a central vein (CV) and a portal vein (PV) connected by a dense plexus, termed the sinusoidal network, which transports metabolite-rich blood from PV to CV. The sinusoidal network contacts each of the hepatocytes, the main parenchymal cell type in the liver, which take up nutrients and toxins from the blood. The liver serves as the central organ of blood detoxification, including the metabolism of medically relevant drugs. Hence, an ongoing

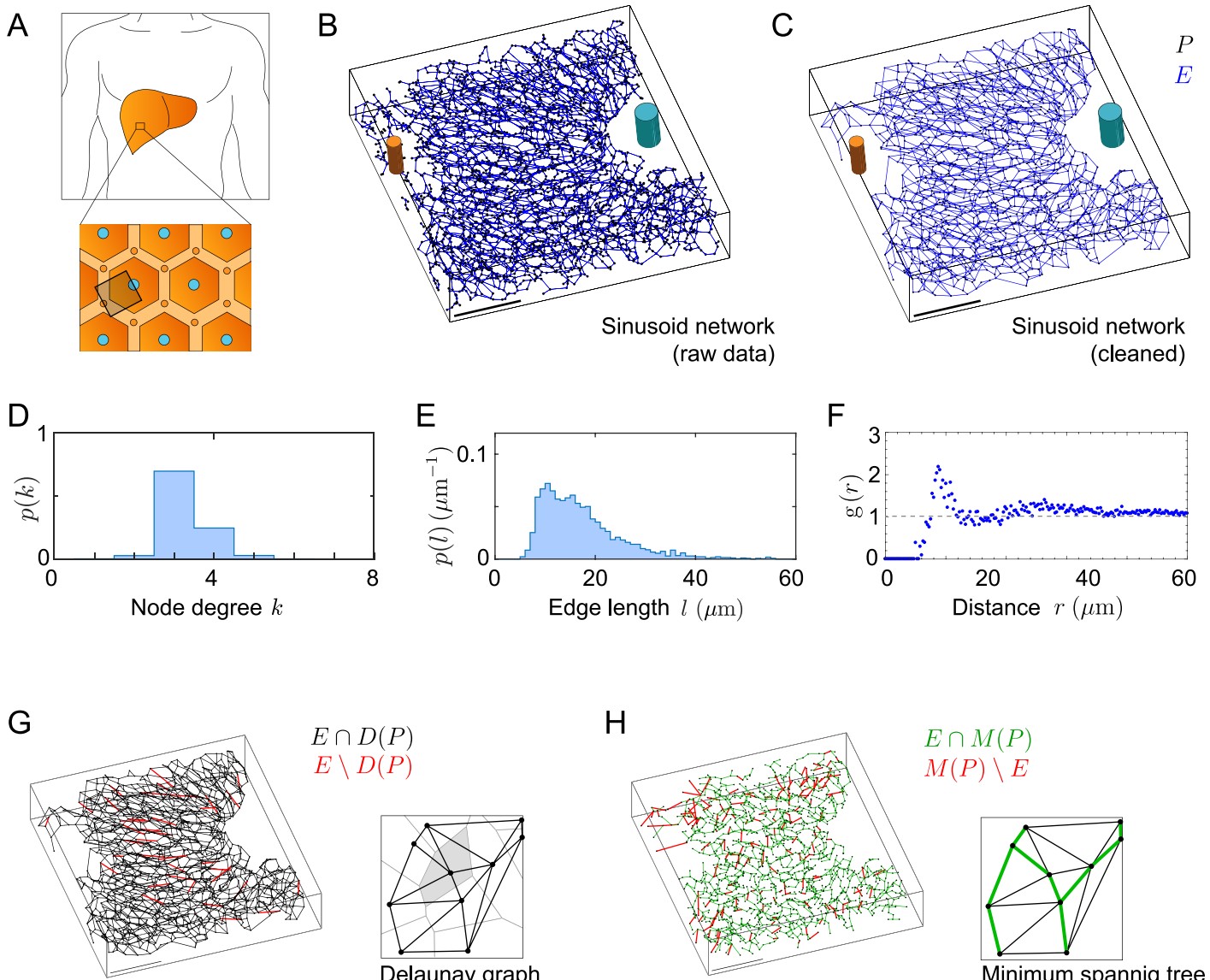

**Fig 1. Sinusoidal networks in liver tissue. A**. The liver is organized into millimeter-sized lobules, each containing a central vein (CV, cyan) and several portal veins (PV, orange). Blood flows from PV to CV through the plexus-like sinusoidal network. **B**. Raw sinusoidal network obtained by high-resolution 3D imaging of mouse liver tissue. Shown is the network skeleton with branch points (black) and edges (blue), defining a spatial network. CV and PV shown schematically. Scale bar: 100 $\mu$m. **C**. Sinusoidal network $\mathcal{N}$ obtained from the raw network by removal of degree-2 nodes and disconnected components, used in all subsequent analysis, see also S1 Movie for a three-dimensional visualization. **D, E, F**. Distribution of node degree $p(k)$, distribution of edge lengths $p(l)$, and radial distribution function $g(r)$ of spatial node positions for the sinusoidal network $\mathcal{N}$, respectively. **G**. The majority of edges of the sinusoidal network are contained in the Delaunay graph corresponding to its set of node positions. Edges of the sinusoidal network not contained in the Delaunay graph are shown in red. Inset: cartoon illustrating a Delaunay graph (black) and Voronoi tessellation (gray) of a set of points. **H**. The sinusoidal network contains the majority of edges of the minimum spanning tree $M(P)$ of its set of node positions $P$. Edges of $M(P)$ not in $E$ are shown in red. Inset: cartoon representation of a minimum spanning tree (green).

intense research effort focuses on the pharmacological modeling of the uptake of drugs from the blood stream in the liver [15–21]. Understanding fluid flow in the sinusoidal network is imperative to improve these models and make functional predictions. Previous simulation studies of blood flow in liver microvasculature were either limited to large veins [17], or considered only small spatial regions [15, 20]. For the entire lobule, spanning from PV to CV, only continuum models had been used to simulate flow [15, 22, 23].

Finally, the sinusoidal network can become damaged upon non-lethal toxification, prompting the question of resilience properties of this network. The liver exhibits remarkable regeneration capabilities. After local damage, sinusoidal networks are expected to self-repair on a time-scale of days or weeks. Yet, on shorter time-scales, the sinusoidal network must cope with any transient reduction in network permeability. By how much the permeability of the sinusoidal network will be reduced after a perturbation is not known.

Here, we analyze network geometry using a digital reconstruction of the sinusoidal network based on high-resolution image data of adult mouse liver [13, 14]. We develop a network generation algorithm that reproduces statistical features of the sinusoidal network (node degree distribution, edge length distribution, mean nematic order parameter), enabling us to simulate arbitrarily sized networks from spatially restricted biological samples and, moreover, to explore *in silico* a design space of three-dimensional networks. While simulating random graphs with given degree distribution is a classical problem of combinatorics [24], and popular software packages exist for common models of random spatial networks [25], we were not aware of previous network generation algorithms that allow to prescribe both degree and edge length distribution.

Sinusoidal networks display a weak nematic alignment along the direction of flow [14, 15, 26], i.e., the edges of the network are not oriented isotropically in all space directions, but exhibit a tendency to be aligned towards a common axis. Using our algorithm, we can systematically vary this nematic alignment in simulated networks. We empirically find a linear relationship between the anisotropic permeability of simulated networks and a nematic order parameter of the networks that quantifies their anisotropic geometry. Permeabilities allow to efficiently compute macroscopic, tissue-level flows using a continuum model [15, 22, 23, 27], thus providing an effective medium theory of fluid transport.

To quantify the fault tolerance of these networks, we introduce a new resilience measure, which we term *permeability-at-risk* and which quantifies changes in network permeability if a given fraction of network links is removed. The resulting permeability-at-risk curves can be considered as a generalization of the bond percolation problem in the theory of random resistor networks [28, 29]. We find that simulated networks with weak nematic order display a substantially increased permeability along the direction of nematic alignment. If the mean nematic order parameter equals that of sinusoidal networks, this increased permeability does not compromise network resilience as compared to isotropic simulated networks. Our minimal transport model, which assumes constant and equal flow resistance per unit length for each edge, predicts that the distribution of computed currents is very inhomogeneous in the network, with a few edges carrying most of the current. This renders these networks highly vulnerable to the removal of high-current edges, despite their resilience against random removal of edges. In the discussion, we speculate on mechanisms such as shear-dependent adaptation of the diameter of sinusoids [30–32], or transient clogging by erythrocytes [33, 34], which would both affect especially high-current edges, and could homogenize the time-averaged distribution of currents in the network, thereby reducing the vulnerability of sinusoidal networks to the removal of high-current edges.

## Results

### Experimental data and network metrics

To analyze the statistical geometry of three-dimensional microvasculature networks, we took advantage of advances in high-resolution imaging of murine liver tissue [13, 14]. Based on segmented three-dimensional image data the skeleton of the hepatic sinusoidal network was computed using MotionTracking image analysis software [13, 35], see Fig 1B.

Next, a cleaned version of the raw network data was computed: (i) small disconnected network components not connected to the largest component were discarded, (ii) connected nodes separated by a distance smaller than a cut-off distance $R_c = 8\ \mu$m (outer diameter of sinusoids) were merged into a single node, (iii) in a subsequent pruning step, dead ends were removed, leaving only nodes with node degree $d \geq 2$. Finally, linear-chain motifs consisting of degree-two nodes in series were replaced by a single link with weight equal to the total length of the linear chain. In rare instances, removal of a linear chain might yield triangles at the extremities of the network, which were also removed. The remaining node points are exactly the branch points of the biological network, whose positions are determined with high precision. This clean-up procedure reduces ambiguity on small network details that were difficult to resolve with current imaging techniques. It provides a faithful representation of the sinusoidal network used in all subsequent analysis, see Fig 1C, as well as S1 Movie.

The skeleton of the sinusoidal network is a spatial network with set of node points $P = \{\mathbf{q}_i\}$ and set of edges $E$ that connect pairs of points. We find that the sinusoidal network is a homogeneous network with mean node degree $\langle d \rangle = 3.3 \pm 0.6$ and a unimodal distribution $p(l)$ of edge lengths $l$ with mean $\langle l \rangle = 17 \pm 8\ \mu$m, see Fig 1C and 1D.

We characterize the relative position of node points $\mathbf{q}_i$ in terms of their normalized radial distribution function

$$g(r) = \frac{1}{\rho_0} \left\langle \int_{|\mathbf{r} - \mathbf{r}_0| = r} d^2\mathbf{r}\, \frac{1}{4\pi r^2} \rho(\mathbf{r} - \mathbf{r}_0) \right\rangle_{\mathbf{r}_0}, \tag{1}$$

where $\rho(\mathbf{r}) = \sum_i \delta(\mathbf{r} - \mathbf{q}_i)$ is the point density of nodes, $\rho_0 = \langle \rho \rangle$ the mean density (with units of an inverse volume), and $\mathbf{r}_0$ the position of a 'central node'. The radial distribution function is closely related to the structure factor, which is used in condensed matter physics to describe the packing of particles and which can be considered the spatial power spectral density of $\rho(\mathbf{r})$ [36]. For an ideal gas, $g(r) = 1$. Fig 1F shows $g(r)$ for the node points of the sinusoidal network. Interestingly, we observe a peak at a characteristic distance $r \approx 10\ \mu$m, indicating a characteristic distance between the branching points of this network. In fact, the resultant $g(r)$ resembles the radial distribution function of a fluid, with short-range repulsion of fluid particles. Interestingly, the observed characteristic distance of branch points corresponds to approximately half the diameter of hepatocytes [14], which is likely to set a characteristic mesh-size of the sinusoidal network. The ratio of branch points to number of hepatocytes is approximately 2 : 1 (i.e., $n = 1643$ branch points and 857 hepatocytes for sample volume shown in Fig 1C).

**The sinusoidal network is a sub-graph of its Delaunay graph.** Given the set $P$ of node positions of the sinusoidal network, we construct the corresponding Delaunay graph with set of edges $D(P)$. The *Delaunay graph* generalizes the familiar Delaunay triangulation in the plane to three space dimensions, and may be considered as the graph that connects 'nearest neighbors' in $P$. Specifically, we may assign to each node $\mathbf{q} \in P$ a neighborhood $V_{\mathbf{q}}$ defined as the set of all points that are closer to $\mathbf{q}$ than to any other point of $P\backslash\mathbf{q}$. Each $V_{\mathbf{q}}$ is a polyhedron. Together, these polyhedra tesselate three-dimensional space, defining the so-called *Voronoi tesselation* of $P$. Now, an edge connecting nodes $\mathbf{q}_1, \mathbf{q}_2 \in P$ belongs to the Delaunay graph $D(P)$ if and only if the corresponding polyhedra $V_{\mathbf{q}_1}$ and $V_{\mathbf{q}_2}$ share a common face. Correspondingly, $D(P)$ is the dual graph of the Voronoi tesselation. The Delaunay graph exists and is unique provided the node points are in general position [37].

Remarkably, we find that the sinusoidal network is a subgraph of the Delaunay graph to very good approximation: 99% of the edges in $E$ are contained in $D(P)$, see Fig 1G. The 1% of edges contained in $E$ but not $D(P)$ (marked red) are longer than average (with a mean length of $57 \pm 19\ \mu$m), yet contribute less than 10% to the network permeabilities computed

below. Thus, the sinusoidal network can be considered a network of nearest-neighbor edges, reflecting the design requirement of a space-filling network that connects all hepatocytes in the tissue.

**The sinusoidal network contains the minimum spanning tree.** The set of node points $P$ defines also a second graph, the minimum spanning tree with edges $M(P)$. The minimum spanning tree is defined as the connected and cycle-free graph with node points $P$ for which the sum of the edge lengths is minimal. Note that the minimum spanning tree $M(P)$ is always a sub-graph of the Delaunay graph $D(P)$ for any set $P$ of points in Euclidean space. Remarkably, the sinusoidal network contains the minimum spanning tree of its node points as a sub-graph to good approximation: 90% of the edges $M(P)$ of the minimum spanning tree belong also to $E$, see Fig 1H. This finding suggests a possible optimization of the sinusoid network for shortest paths.

## A network generation algorithm for spatial networks

We developed a Monte-Carlo algorithm to generate synthetic networks that faithfully reproduce the statistical features of spatially restricted samples of hepatic sinusoidal networks, using multi-objective optimization of both node degree and edge length distribution, see Fig 2A.

Our algorithm proceeds in two steps: first, we simulate surrogate node positions $P_{\text{sim}}$, followed by a second step of selecting a set of edges $E_{\text{sim}}$ connecting these nodes. Importantly, our analysis of sinusoidal network graphs allows us to restrict to simulated networks that are subgraphs of the Delaunay graph $D(P_{\text{sim}})$ and at the same time contain the minimum spanning tree $M(P_{\text{sim}})$, $M(P_{\text{sim}}) \subseteq E_{\text{sim}} \subseteq D(P_{\text{sim}})$. This restriction dramatically reduces the search space without which network optimization would be computationally unfeasible.

In the first step of the algorithm, random node points are generated by simulating random packings of equally sized hard spheres. This minimal model comprises only two fit parameters (the volume fraction of spheres and their radius), and reproduces the characteristic feature of short-range repulsion in the radial distribution function $g(r)$ for the center positions of the spheres, see Fig 2B. The final set of node positions $P_{\text{sim}}$ is determined by taking a random selection of about 20% of simulated hard spheres centers, to match the measured density $\rho_0$ of node positions in sinusoidal networks. This random selection does not change $g(r)$, as it scales down both denominator $\rho_0$ and numerator $\langle \int d\mathbf{x}\, \rho \rangle$ in Eq (1) by the same factor. This random selection reflects volume exclusion by hepatocyte cells (which comprise a volume fraction of about 80% in liver tissue [7]) and other cell types.

In the second step, we select a subgraph of the Delaunay graph $D(P_{\text{sim}})$ of the set of surrogate node positions $P_{\text{sim}}$, using multi-objective optimization for node degree and edge length distribution. We introduce the cumulative probability density functions (CDF) $\text{CDF}(d_0) = \sum_{d \leq d_0} p(d)$ and $\text{CDF}(l_0) = \int_0^{l_0} dl\, p(l)$ for node degree and edge length distribution of the sinusoidal network, respectively, as well as analogous CDFs $\text{CDF}_{\text{sim}}(d)$ and $\text{CDF}_{\text{sim}}(l)$ for simulated networks. Note that $p(l)$ has dimensions of an inverse length, hence $\text{CDF}(l)$ is dimensionless. We define two cost functions $C_d$ and $C_e$ for the node degree and edge length distribution, respectively, as the difference of the CDFs

$$C_d = \sum_{d=1}^{\infty} |\text{CDF}(d) - \text{CDF}_{\text{sim}}(d)|^2 \tag{2}$$

and

$$C_e = \frac{1}{R_c} \int_0^{\infty} dl\, |\text{CDF}(l) - \text{CDF}_{\text{sim}}(l)|^2 \tag{3}$$

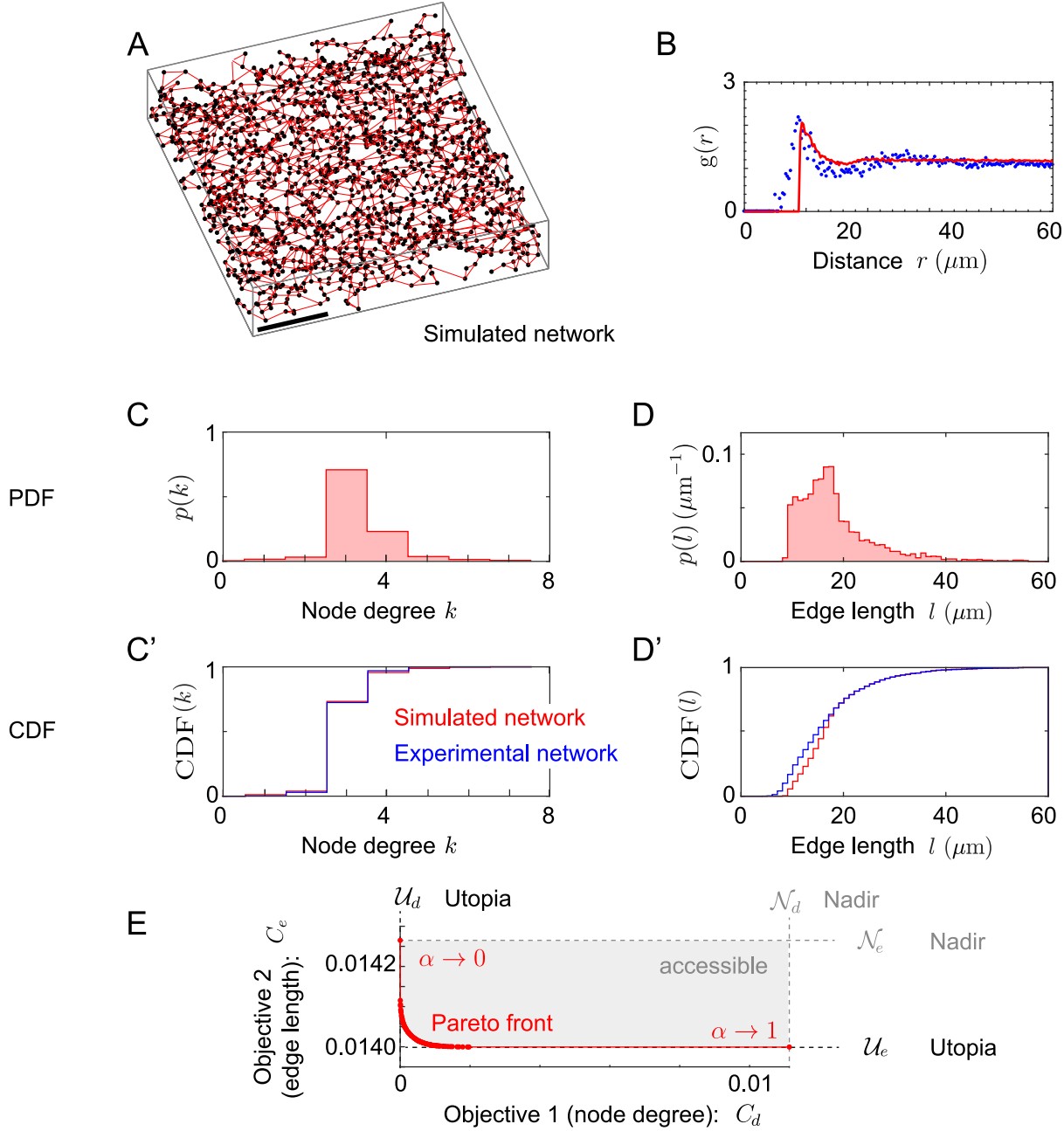

**Fig 2. Simulation of networks with prescribed statistics. A**. Example of simulated network resulting from multi-objective optimization. The size of the simulated domain matches the dimensions of the experimental network samples (420 $\mu$m × 450 $\mu$m × 90 $\mu$m). We used Monte-Carlo simulations of a packing of hard spheres to determine node positions. **B**. Comparison of radial distribution function $g(r)$ from sinusoidal network (black dots), and simulated node positions (red). **C, C'**. Probability density function (PDF) and cumulative distribution function (CDF) for the node degree distribution $p(k)$ of a simulated network (red), and the sinusoidal network (blue). **D, D'**. Same as panel C, but for the edge length distribution $p(l)$. **E**. Pareto-front of multi-objective optimization. Shown are normalized costs for each of the two objectives: optimization of node-degree and optimization of edge-length distributions, for varying weight $\alpha$ of the multi-objective cost function.

We normalized $C_e$, using the cut-off distance $R_c$ as a characteristic length scale. As illustration, Fig 2C and 2D shows probability distribution functions (PDF) and cumulative distributive functions (CDF) for the simulated network from panel A. The difference in CDFs provides a robust distance measure, which is particularly robust against small shifts of the

probability distributions $p(d)$ and $p(l)$. We also introduce a multi-objective cost function as weighted mean of the individual cost functions, parametrized by a weight $\alpha$

$$C_\alpha = \alpha C_d + (1 - \alpha) C_e, \tag{4}$$

We determined sets of edges $E^*_{\mathrm{sim}}(\alpha)$ that minimize $C_\alpha$ for given $\alpha$ by simulated annealing, see Methods section for details. Note that choosing either $\alpha = 0$ or $\alpha = 1$ would correspond to optimizing only a single objective, $C_e$ or $C_d$, respectively. These single-objective optima set the *utopia points*, i.e., the best-achievable values for each individual cost function, $\mathcal{U}_d = C_1[E^*_{\mathrm{sim}}(1)] = C_d[E^*_{\mathrm{sim}}(1)]$, and $\mathcal{U}_e = C_0[E^*_{\mathrm{sim}}(0)] = C_l[E^*_{\mathrm{sim}}(0)]$, see Fig 2E. The value of the respective other cost function define the *nadir points* $\mathcal{N}_e$ and $\mathcal{N}_d$. The definition of the nadir points requires to take a limit of $\alpha$, $\alpha \nearrow 1$ or $\alpha \searrow 0$, as $\mathcal{N}_e = \lim_{\alpha \nearrow 1} C_e[E^*_{\mathrm{sim}}(\alpha)]$, and $\mathcal{N}_d = \lim_{\alpha \searrow 0} C_d[E^*_{\mathrm{sim}}(\alpha)]$. Otherwise, the values of $C_e$ and $C_d$ would not be well-defined, because single-objective optimization with $\alpha = 1$ or $\alpha = 0$ would yield multiple optimal networks with different values for the respective other objective.

For multi-objective optimization with $0 \leq \alpha \leq 1$, the values of the individual cost functions for the optimal network are bounded by the utopia and nadir points, $\mathcal{U}_d \leq C_d[E^*_{\mathrm{sim}}(\alpha)] \leq \mathcal{N}_d$, and $\mathcal{U}_e \leq C_e[E^*_{\mathrm{sim}}(\alpha)] \leq \mathcal{N}_e$. The curve $(C_d[E^*_{\mathrm{sim}}(\alpha)], C_e[E^*_{\mathrm{sim}}(\alpha)])$, $0 \leq \alpha \leq 1$, defines a *Pareto front* that separates a region of impossible pairs of $(C_d, C_e)$ values, from a region of possible values. Remarkably, this Pareto front deviates only little from the straight lines defined by $C_d = \mathcal{U}_d$ and $C_e = \mathcal{U}_e$ for intermediate values of $\alpha$. Hence multi-objective optimization for both degree and edge length distribution can be achieved with minimal impairment on the individual cost functions. We thus find *a posteriori* that our algorithm is robust with respect to the particular choice of $\alpha$. In the following, we use the value $\alpha^* = (\mathcal{N}_e - \mathcal{U}_e)/(\mathcal{N}_d - \mathcal{U}_d + \mathcal{N}_e - \mathcal{U}_e)$, which corresponds to choosing equal weights for the individual cost functions $C_d$ and $C_e$ after these have been normalized such that their utopia and nadir points are mapped to zero and one, respectively, see Methods section for details. This normalization requires in particular knowledge of the nadir points, which can only be computed as limits of the Pareto front, but not using single-objective optimization alone.

We analyzed $n = 3$ sinusoidal networks in total, each with approximate sampling volume $420\ \mu m \times 450\ \mu m \times 90\ \mu m$ and 1643, 2689, 1996 nodes, respectively, which all gave similar results.

The simulated networks offer a three-fold advantage compared to the spatially restricted experimental samples: (i) they allow to analyze transport properties in larger networks (here: about 2.6-fold more nodes compared to central regions of interest), thus reducing statistical fluctuations arising from small sample volumes (see S1 Appendix, Fig S1), (ii) they allow to analyze networks with larger spatial extent in $z$ direction, thus reducing boundary effects in flow simulations (see also S1 Appendix, Fig S2), and (iii) they allow to change nematic order of the network by an external control parameter, as considered next.

## Nematic order determines anisotropy of network permeability

The sinusoidal network is not isotropic, but displays weak nematic alignment along a direction parallel to the direction of blood flow along the PV-CV axis, see Fig 3A. *Nematic order* describes the partial alignment of local axes, such as direction axes of anisotropic objects. This concept was originally introduced to describe the mutual alignment of elongated molecules in liquid crystals [38], yet has been extended to characterize also mutual alignment of biological structures, such as cytoskeletal filaments, or polarized cells in a tissue [39–41]. Technically, one distinguishes polar order (parallel alignment of single-headed vectors ↑↑, but not anti-

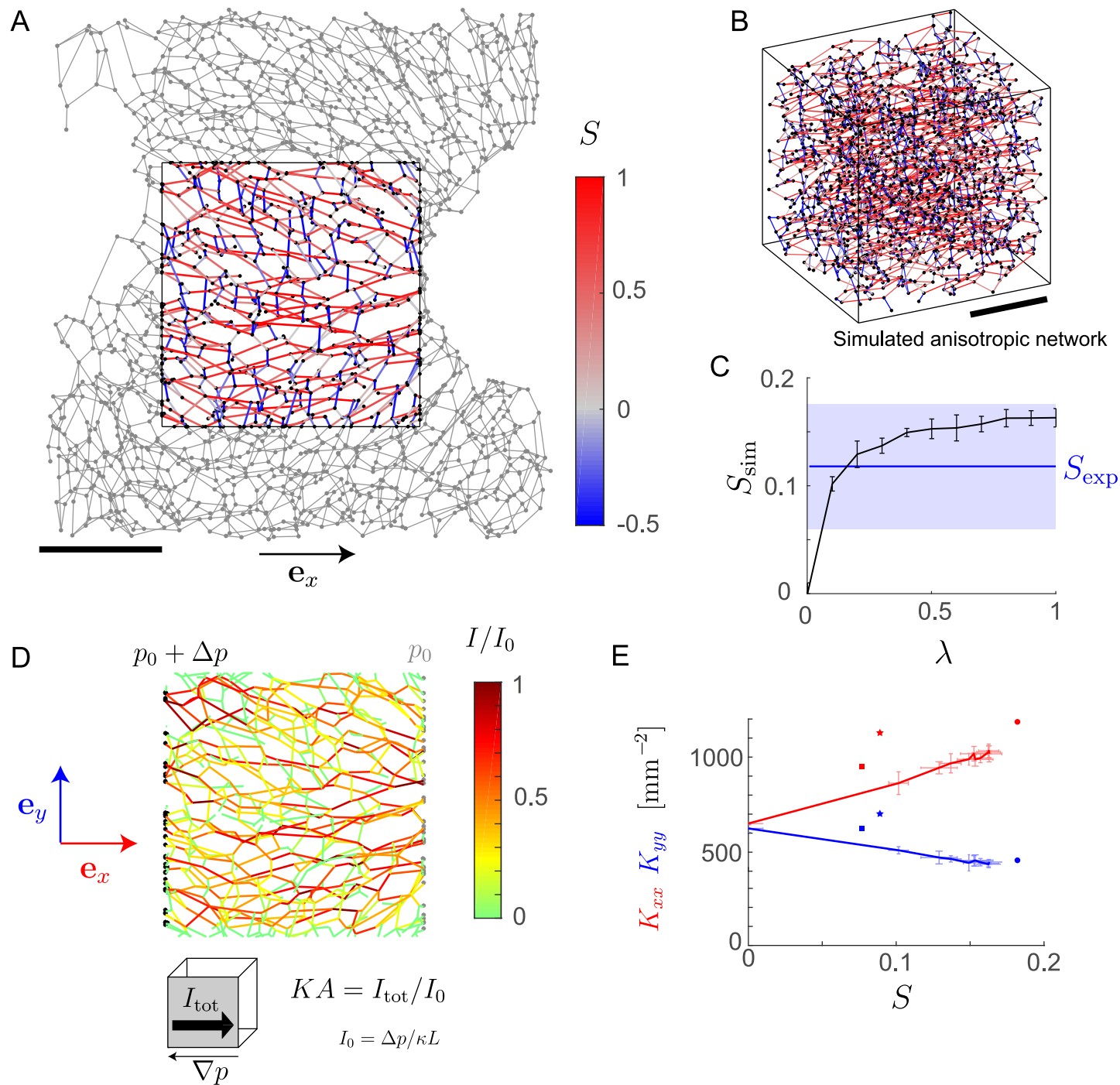

**Fig 3. Nematic order determines anisotropy of network permeability. A**. Sinusoidal network with edges inside a central region of interest color-coded according to their parallel alignment with the CV-PV axis $\mathbf{e}_x$ (red: parallel alignment, blue: perpendicular alignment). Scale bar: 100 $\mu$m. **B**. Simulated network with an external alignment field imposed during optimization ($\lambda = 1$). Color-code as in panel A. **C**. Nematic order parameter $S_{\text{sim}}$ of simulated networks as function of external alignment field strength $\lambda$ ($n = 3$ realizations, mean±s.e.). For comparison, the nematic order parameter $S_{\text{exp}}$ of sinusoidal networks measured along the CV-PV axis is shown (blue region represents mean±s.e., $n = 3$). **D**. Computed flow through the sinusoidal network for imposed pressure difference at opposing boundary surfaces. **E**. Computed permeabilities of simulated networks (solid lines) as function of the nematic order parameter $S$ (permeability $K_{xx}$ along $\mathbf{e}_x$ parallel to the direction of nematic order: red, permeability $K_{yy}$ along $\mathbf{e}_y$ perpendicular to the direction of nematic order: blue, mean±s.e., $n = 3$ realizations). Symbols indicate the respective permeability sinusoidal networks ($n = 3$; filled circle correspond to network shown in panel A).

parallel alignment ↑↓), and nematic alignment (parallel alignment of undirected axes, ||). The local mean direction of nematic alignment is called the *nematic director*.

Previous work on liver tissue has shown that the local director of sinusoidal network anisotropy follows curvilinear flow lines, where PV and CV serve as source and sink, respectively [14, 26]. Inside a central region of the lobule, these flow lines are approximately straight, which simplifies our subsequent analysis. We quantify nematic alignment of the sinusoidal network inside this central region in terms of a *nematic order parameter*. This nematic order parameter is computed as the statistical average of the alignment of individual edges with the reference direction

$$S = \left\langle \frac{3}{2} (\mathbf{e}_j \cdot \mathbf{e}_x)^2 - \frac{1}{2} \right\rangle,$$ (5)

Here, $\mathbf{e}_x$ is a unit vector pointing along the CV-PV axis (which is parallel to the $x$ axis for the given data set and sets the reference direction), and $\mathbf{e}_j$ is a unit vector parallel to the $j$-th edge. In Eq (5), averaging is performed over all edges completely located inside the region of interest. To compute $S$, we choose a cuboid region of interest of dimensions $L_x \times L_y \times L_z$, located in the central region of the lobule, with cuboid edges aligned with the $x$, $y$, $z$ axes of a coordinate system, see Fig 3A. We found $S = 0.12 \pm 0.06$ (mean±s.e., $n = 3$ independent data sets from different mice; $S = 0.18$ for the data set in Fig 3A). Note that in Eq (5), the contribution of each edge is independent of its length. Since edge lengths are distributed rather homogeneously, such weighting would not change numerical results. However, a weighting by edge length would introduce a spurious bias towards longer edges in simulations of anisotropic networks using Eq (6).

Next, we modified our Monte Carlo network generation algorithm, to generate networks that likewise show nematic alignment along a specified axis. For that aim, we compute a nematic order parameter $S_{\text{sim}}$ for simulated networks analogous to Eq (5) and use an augmented cost function in the optimization

$$C_S = C_\alpha - \lambda \, v \, S_{\text{sim}},$$ (6)

with interaction parameter λ. Here, $v = \alpha^*(\mathcal{N}_d - \mathcal{U}_d) + (1 - \alpha^*)(\mathcal{N}_e - \mathcal{U}_e)$ is a normalization constant introduced for numerical convenience, corresponding to the use of a normalized cost function $\bar{C}$, see Eq (10). For simplicity, we do not account for a possible dependence of edge lengths on their orientation in space. Fig 3B shows an example of a simulated anisotropic network (with nematic order parameter $S_{\text{sim}} \approx 0.16$, similar to the value $S \approx 0.18$ measured for the experimental sinusoidal network shown in Fig 3A). More generally, the nematic order parameter $S_{\text{sim}}$ is a monotonic increasing function of the interaction parameter λ in Eq (6), see Fig 3C. The nematic order parameter $S_{\text{sim}}$ of simulated networks saturates at a maximal value $S_{\text{sim}} = 0.16 \pm 0.01$, suggesting that there exists a maximal value of $S$ compatible with the given degree and edge length distribution.

The geometric anisotropy of the sinusoidal network results in anisotropic transport properties. We will quantify these by an anisotropic permeability, which represents a size-independent "material property" of the spatial network. This will facilitate the comparison to simulated networks, as opposed to flow computations for physiological boundary conditions with influx and outflux at the central and portal vein, respectively. We use a minimal flow model that assumes a constant resistance $\kappa$ per unit length for each edge. This assumption is justified since Reynolds numbers are small [15]. Thus, we can model flow using Kirchhoff's

laws

$$(p_j - p_k) \quad = \kappa \, l_{j,k} \, J_{j,k} \text{ for all edges connecting nodes } j \text{ and } k \text{ (Ohm's law) ,} \tag{7}$$

$$0 \quad = \sum_k J_{j,k} \text{ for all nodes } j \text{ (conservation of current).} \tag{8}$$

Here, $J_{j,k}$ denotes the signed current through the directed edge connecting node $j$ and $k$ (with units $m^3 s^{-1}$), $l_{j,k}$ the length of that edge, and $\kappa$ a constant resistance per unit length (with unit $Pa \, m^{-4} s^{-1}$), corresponding to the simplifying assumption of a constant diameter of sinusoid tubes.

Fig 3D shows the flow computed for the chosen region of interest of the sinusoidal network. Here, we impose a pressure difference $\Delta p$ at opposing boundaries of the region of interest as indicated. The rationale is that both the direction of mean flow from previous continuum models of flow [15], as well as the local nematic direction of the sinusoidal network are approximately uniform in the central region of the liver lobule, parallel to the CV-PV axis. For the chosen region of interest, these directions are thus expected to be perpendicular to region boundaries with the imposed pressure difference. Specifically, we identify all nodes close to the two boundary surfaces of the cuboid region normal to the $x$ axis as either sink and source nodes, with $p_i = p_0$ and $p_i = p_0 + \Delta p$, respectively. Solving the linear problem defined by these boundary conditions and Eq (7), with conservation of current at all nodes that are neither source nor sink, yields the pressure at these remaining nodes and the currents $J_{j,k}$. The total inflow $J_x$ at the source nodes equals the total outflow at the sink nodes. Contrary to intuition, we find a distribution of flows that is very inhomogeneous, see Fig 3D. As an interesting side remark, also an inhomogeneous distribution of flows can allow for a homogeneous supply of the tissue, see S1 Appendix, Fig S3, for results from a minimal model of metabolic uptake from the sinusoidal network by surrounding tissue.

The *permeability* of a network in $x$ direction is proportional to the ratio of the total current $J_x$ divided by the imposed pressure difference $\Delta p$. We normalize the permeability by the dimension $L_x$ of the cuboid in $x$ direction, its cross-sectional area $A_x = L_y L_z$, and the resistance per unit length $\kappa$ of individual edges, to obtain a normalized permeability (with units of an area density) as

$$K_{xx} = \frac{L_x}{A_x} \frac{\kappa J_x}{\Delta p}. \tag{9}$$

Analogously, we define $K_{yy}$ and $K_{zz}$ for the $y$ and $z$ direction, respectively. This definition of a normalized permeability defines a purely geometrical measure of the network that is independent of $\kappa$, a property known as Darcy's law [42]. We confirmed that indeed Darcy's law holds approximately for the large samples used, i.e., the normalized permeability is indeed independent of the dimensions of the cuboid if boundary effects can be neglected, see S1 Appendix, Fig S1. For smaller sample volumes, computed permeabilities display a high statistical variability, but still have the same mean value. To convey the geometric meaning of the normalized permeability, we consider a minimal network that consists of straight lines parallel to the $x$ axis, running from one boundary face of the cuboid to the opposing face: there, we have $K_{xx} = K_0$, where $K_0$ is the area density of these lines in any cross section of the cuboid.

Fig 3E shows computed normalized permeabilities $K_{xx}$ and $K_{yy}$ of sinusoidal networks in the direction of nematic alignment and normal to it, respectively. We have $K_{xx} = 1.1 \pm 0.1 \cdot 10^3$ $mm^{-2}$ and $K_{yy} = 0.6 \pm 0.1 \cdot 10^3 \, mm^{-2}$ (mean±s.e., $n = 3$). The computed permeability in $z$ direction, $K_{zz} = 0.9 \pm 0.3 \cdot 10^3 \, mm^{-2}$, is rather unreliable due to the small dimension $L_z$ of the

experimental sample in $z$ direction. The anisotropy ratio $K_{xx}/K_{yy} \approx 1.83$ is consistent with a previously reported value 2.2 computed for a 0.15 $\mu$m $\times$ 0.15 $\mu$m $\times$ 0.15 $\mu$m sample of the sinusoidal network imaged using $\mu$CT [15].

For the simulated networks, we find that the permeability $K_{xx}$ in the direction of nematic alignment increases linearly as a function of the nematic order parameter $S_{sim}$, while the permeability $K_{yy}$ in normal direction decreases, see Fig 3E.

We can estimate the resistance per unit length $\kappa$ of sinusoids using the Hagen-Poiseuille formula as $\kappa = 8\mu/(\pi R^4) \approx 1.1 \cdot 10^{20}$ Pa m$^{-4}$ s, where $\mu = 3.5 \cdot 10^{-3}$ Pa s is the dynamic viscosity of blood [19], $2R = 6$ $\mu$m the inner diameter of sinusoids. Given a typical pressure difference between portal and central vein $\Delta p = 100 - 250$ Pa (1 – 2 mm Hg) [15, 18], and a typical spacing of $L = 0.5$ mm, this value implies a volumetric flow rate of $J = 2 - 5 \cdot 10^{-9}$ ml s$^{-1}$ through a single sinusoid connecting portal and central vein with maximal flow velocity of $v = 0.13 - 0.32$ mm s$^{-1}$, which validates our low Reynolds number approximation (Re $\approx 10^{-4}$). The normalized permeabilities computed here correspond to non-normalized permeabilities $\mu K_{xx}/\kappa = 3.5 \pm 0.3 \cdot 10^{-14}$ m$^2$ and $\mu K_{yy}/\kappa = 1.9 \pm 0.3 \cdot 10^{-14}$ m$^2$ with units of an area as commonly used in the theory of porous media. These values agree within a factor of two with permeabilities along the radial and circumferential direction of the lobule computed previously for a smaller sample volume, $1.56 \cdot 10^{-14}$ m$^2$ and $1.76 \cdot 10^{-14}$ m$^2$, respectively [15]. Note that the resistance $\kappa$ is very sensitive to the assumed diameter of sinusoids; a 10% decrease in $R$ would decrease the non-normalized permeability by a factor of two.

## Permeability-at-risk

We used the minimal flow model to study the resilience properties of sinusoidal networks, i.e. the ability of the network to support flow even after partial damage. If edges are removed from a network, the permeability of the network decreases. This known fact is easily proven by adding an edge and noting that for constant net flux $J$ the pressure difference $\Delta p$ must drop due to re-routing of flow in accordance to Helmholtz' theorem on the minimization of energy dissipation of low-Reynolds number flows [43]. We quantify network resilience in terms of *permeability-at-risk* curves, see Fig 4A. Specifically, we plot the permeability of perturbed networks as a function of the fraction $\gamma$ of removed edges. At a critical value $\gamma^*$ of $\gamma$, this permeability becomes zero, indicating the *percolation threshold* of the networks. We consider two different scenarios for the removal of edges: (i) removal of high-current edges, i.e., those edges that carried the highest current in the unperturbed network, (ii) random removal of edges with probability proportional to their length. We find that removing high-current edges results in a much steeper decrease of the permeability (scenario i, solid line), as compared to a random removal of edges (scenario ii, dashed). This shows that the edges that carry a high current in the unperturbed network are indeed crucial for the global transport properties of the network. Transport by these high-current edges is only partially compensated by re-routing of flow through low-current edges. In the absence of additional perturbations, low-current edges are dispensable and the network permeability decreases only little if the majority of low-current edges is removed (e.g. if all edges with current below the 25% percentile among those edges that do not touch the boundary of the region of interest are removed, $K_{xx}$ decreases by 11.1 $\pm$ 5.1%). However, a network without these low-current edges displays permeability-at-risk curves that decay even steeper as function of $\gamma$, see S1 Appendix, Fig S2: low-current edges contribute at least partially to network resilience.

Remarkably, the computed permeability-at-risk curves match to very good extent in both scenarios for sinusoidal networks and simulated networks. Here, the nematic order parameter $S$ of simulated networks matches the value of the experimental sinusoidal networks.

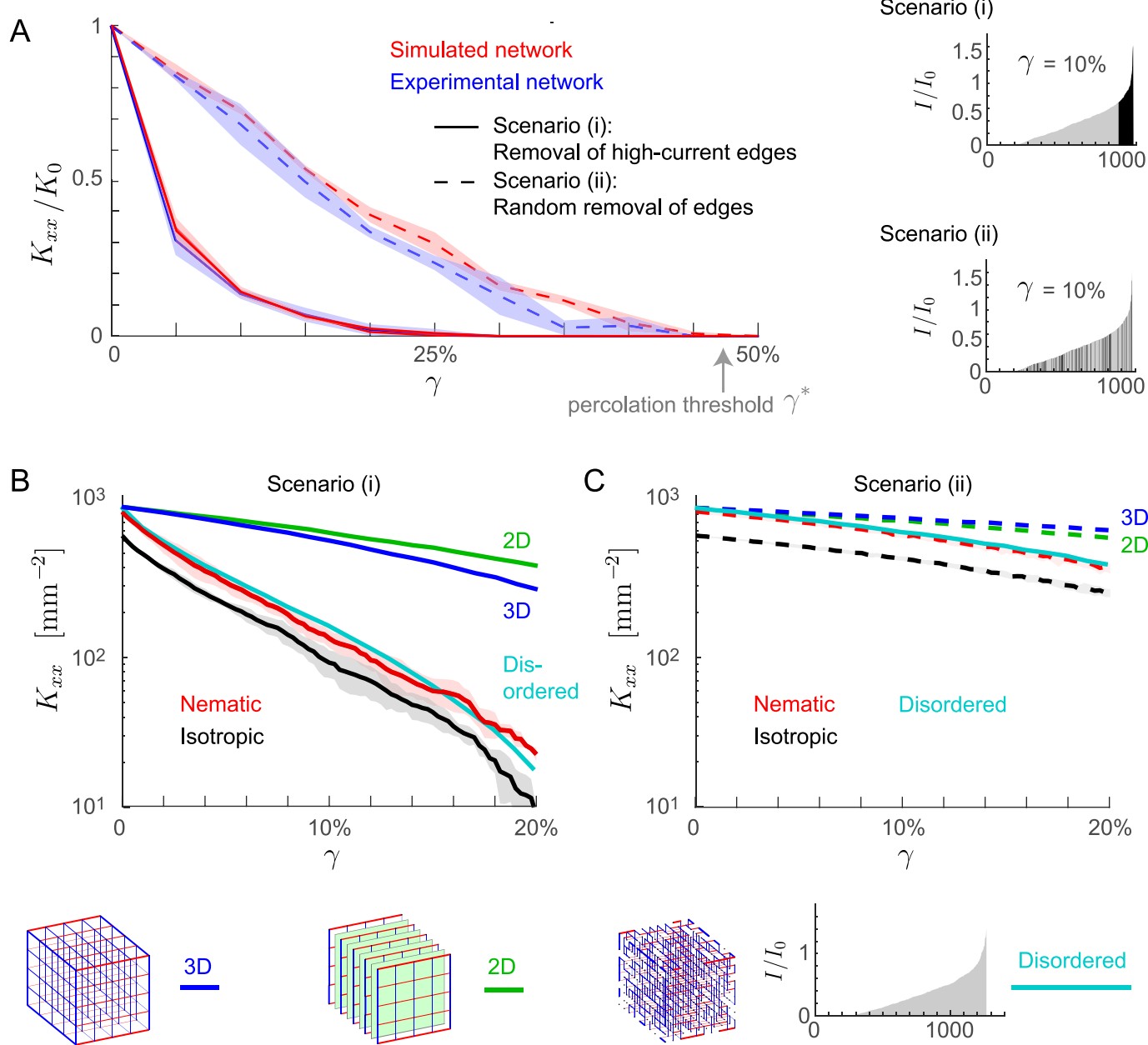

**Fig 4. Resilience of simulated three-dimensional transport networks. A**. Permeability-at-risk. Shown is the permeability of both experimental and simulated networks after perturbation as a function of the fraction $\gamma$ of removed edges for two basic scenarios: removal of edges that carried the highest current in the unperturbed network (solid), and random removal of edges with removal probability proportional to their length (dashed). Results are similar for sinusoidal networks (blue) and simulated networks (red). Above a critical $\gamma^*$ (percolation threshold), the permeability drops to zero. Permeabilities $K_{xx}$ were computed along the direction of nematic order ($x$ direction), and normalized by the permeability $K_0$ of the unperturbed network to facilitate comparison ($K_0 = 943 \pm 121$ mm$^{-2}$ for sinusoidal networks, $K_0 = 943 \pm 32$ mm$^{-2}$ for simulated networks). Shaded areas indicate mean ± s.e. ($n = 3$ networks for each condition; region of interest for sinusoidal networks as in Fig 3A). **B**. Logarithmic plot of permeability as function of $\gamma$ for simulated networks for scenario *i* of removal of high-current edges for isotropic networks (black; $\lambda = 0$), and nematic networks (red; $\lambda = 0.2$, resulting in $S \approx 0.129$, comparable to the value of sinusoidal networks). We compared the permeability-at-risk curve of simulated networks to that of three simple networks: full cubic lattice (blue; $5 \times 5 \times 5$, lattice spacing 31.6 $\mu$m ensuring $K_{xx}(\gamma = 0) = 10^3$ mm$^{-2}$), layers of square lattices (green; $5 \times (5 \times 5)$, lattice spacing 31.6 $\mu$m), random sub-lattice of cubic lattice (cyan; sub-lattice of $10 \times 10 \times 10$, with filling fraction 48%, mean of $n = 10$ realizations, lattice spacing 14.3 $\mu$m). Cartoons of the three simple networks are shown below. **C**. Analogous to panel B for scenario *ii* of random removal of edges.

Even for a strong reduction of network permeability, the majority of network nodes can still be connected to both the source and the sink. For example, we find for $\gamma = 10\%$ that only $1.3 \pm 2.1\%$ and $0.5 \pm 0.1\%$ of nodes in the simulated network are disconnected from source or sink in scenario *i* and *ii*, respectively. As expected, these fractions increase to 100% if $\gamma$ approaches the respective percolation thresholds.

Next, we asked how nematic order of a network influences its resilience, using our network generation algorithm that reproduces the statistical features of sinusoidal networks, while allowing us to tune its nematic order parameter. We find that nematic simulated networks (whose nematic order parameter matches those of the sinusoidal networks) have a higher permeability along the direction of nematic alignment than isotropic networks, not only in the absence of perturbations, but also if edges are removed, see Fig 4B and 4C.

Nonetheless, in scenario *i*, where high-current edges are removed first, both sinusoidal networks and simulated networks display a comparatively low resilience. To understand the origin of this vulnerability, we consider simple example networks. In Fig 4B and 4C, we plot resilience curves of simple regular lattices: a full cubic lattice (blue) and a layered stack of square lattices (green). Both regular lattices display a much weaker reduction of network permeability upon edge removal. We attribute this apparent resilience of regular lattices to the fact that a large subset of their edges carry the same current in the unperturbed network, and thus can be considered equally important. In contrast, disordered networks display the same vulnerability: we tested random sub-lattices of a cubic lattice and observed a similar dramatic drop in network permeability if high-current edges are removed, see Fig 4B and 4C (cyan). The filling fraction of 48% of this sub-lattice served as fitting parameter that allowed to change the disorder of the network in a continuous manner. In conclusion, a low resilience against removal of high-current edges, as observed in sinusoidal networks, appears to be a general property of disordered networks, which are typically characterized by an inhomogeneous distribution of currents across their edges.

## Discussion

We analyzed geometry and transport properties of microvasculatory sinusoidal networks in three-dimensional liver tissue. We took advantage of recent advances in three-dimensional imaging and digital reconstruction of liver tissue [13, 14]. These allowed us to extend previous work, which predominantly addressed two-dimensional biological transport networks [1–3, 44, 45], to three space dimensions. To overcome the limitations of spatially restricted samples of sinusoidal networks, we developed a Monte-Carlo algorithm for multi-objective optimization to generate simulated networks that faithfully reproduce the statistical features of digitally reconstructed sinusoidal networks (degree and edge length distribution). This algorithm is robust and does not depend on how the individual objectives are weighted, yet would fail if only a single objective were used. By varying the nematic alignment of simulated networks, we quantified the relationship between the nematic order parameter and the anisotropy of network permeability: for increasing nematic order, the permeability increases along the direction of nematic alignment, yet decreases in the perpendicular directions, displaying a linear dependence on the nematic order parameter. We find that the nematic order parameter of simulated networks is bounded by a maximal value $S_{\max} \approx 0.19$ if we simultaneously prescribe other statistical quantities of sinusoidal networks (i.e., total number of node points and edges, radial distribution function $g(r)$ of node points, node degree distribution $p(d)$, and edge length distribution $p(l)$). This maximal value is close to the maximal nematic order parameter of the three sinusoidal networks analyzed ($S_{\exp} = 0.182, 0.077, 0.089$).

To characterize the resilience of both sinusoidal and simulated networks in a scale-invariant manner, we introduced the notion of *permeability-at-risk*, i.e., we compute the residual permeability of the network as a function of the fraction $\gamma$ of edges removed. This measure is more general than the familiar percolation threshold at which the permeability becomes zero almost certainly, or the *fault tolerance* considered in Tero *et al.* [2], which quantifies the probability of percolation upon removal of an individual edge. The permeability-at-risk curves for different networks follow a characteristic, concave shape, monotonically decaying as function of $\gamma$ and becoming zero at the percolation threshold of the network. Yet, individual curves do not collapse on a master curve, but reflect properties of individual networks. Computed permeability-at-risk curves match for both sinusoidal and simulated networks. This serves as additional validation of our network generation algorithm. Using simulated networks that mimic real sinusoidal networks, we were able to vary the nematic order parameter $S_{sim}$ of the simulated networks in the range $0 \leq S_{sim} \leq S_{max}$. We found that networks with higher nematic order not only have a higher permeability in the absence of perturbations as compared to isotropic networks with $S = 0$, but that the nematic networks retain this property even if edges are removed. Thus, these nematic networks optimize performance without compromising resilience. Nonetheless, according to our minimal transport model, sinusoidal networks and their simulated counterparts are rather vulnerable to the removal of high-current edges. As a benchmark, we compared permeability-at-risk curves of sinusoidal networks to those of both regular lattices, as an example of ordered networks, as well as sub-lattices of such regular lattices, as an example of disordered networks. We find that the regular lattices exhibit higher resilience, whereas disordered networks have similar permeability-at-risk curves as the sinusoidal network. We attribute this lower resilience of disordered lattices to the presence of a small fraction of edges that carry substantially higher currents. Indeed, we observed an inhomogeneous distribution of flows also in the disordered network, but not in the regular networks. We note that the sub-lattice represents an example of a random resistor network, for which a rich body of theoretical results exist [28, 29].

As a general observation, we found that disordered networks with higher nematic order parameter *S* not only have a higher permeability in the absence of perturbation, but also display a higher permeability-at-risk, provided the total length of the network is kept fixed. Thus, for disordered networks of constant total length, there is no trade-off between permeability and resilience. S1 Appendix, Fig S5, illustrates this effect, using a minimal model of disordered networks given by sub-lattices of a regular cubic lattice. In contrast, adding redundant edges perpendicular to the mean flow direction increases network resilience for a given number of removed edges.

Previous studies on optimal transport networks emphasized building and maintenance costs of networks, which were modeled as functions of total network length [9]. The multi-objective optimization used here to simulate synthetic networks adjusts both node degree and edge length distribution and thus controls the total length of the network (by fixing the total number of edges and their mean length). In the case of sinusoidal networks, however, we anticipate that other design constraints are even more important than building costs. A design principle of sinusoidal networks may indeed be the addition of redundant edges transversal to the flow direction, in addition to a backbone of edges along the mean direction of flow.

To compute network permeabilities, we used a minimal transport model based on linear Kirchoff equations with constant resistance per unit length. We now discuss limitations to this model. Generally, the apparent viscosity of blood flow depends on vessel diameter by the Fahraeus-Lindquist effect [46, 47]. In our case, however, the variation of sinusoid diameter in the sinusoidal network is small [13], which validates our assumption of a constant viscosity. The assumed linear relation between currents and pressure differences is valid for Newtonian fluids

at low Reynolds numbers. However, blood is a non-Newtonian, shear-thinning fluid, whose apparent viscosity decreases with flow [48–50]. This implies a reduced effective resistance at higher flow rates, which would favor an even more inhomogeneous distribution of currents within the network as reported here. As a second effect, at branch points of the network, the daughter branch with the higher flow will receive on average a fraction of red blood cells that is higher than the proportional share [51]. This so-called phase-separation or network Fahraeus effect could result in an even more inhomogeneous distribution of hematocrit.

Under the simplifying assumption that all sinusoid tubes have a constant and equal flow resistance, we predict that the distribution of currents in the network is very inhomogeneous, with a small number of edges carrying a substantial part of the flow. While similar observations had been made previously for hierarchical networks [52, 53], we had expected a more homogeneous distribution in the plexus-like sinusoidal network due to its high number of loops, and because hepatocytes require a homogeneous supply with oxygen and nutrients throughout the liver lobule.

We speculate that adaptive mechanisms, like shear-dependent regulation of sinusoid diameter [30–32], or sinusoid contraction dependent on local concentration of solutes [54], may facilitate a more homogeneous distribution. Moreover, the diameter of red blood cells is only slightly smaller than the diameter of the sinusoids, which can result in transient clogging of sinusoids, especially where flow is high [33, 34, 55] or where the diameter of sinusoids is reduced by adaptive mechanisms. We propose that transient clogging (corresponding to a time-varying network [56]), can likewise result in a more homogeneous time-averaged flow distribution. Notwithstanding these uncertainties, our simple model serves as first approximation, which provides a suitable tool to characterize the geometry of spatial networks, even if actual rates of blood flow in liver tissue should differ.

In addition to the sinusoidal network, liver tissue comprises a second network, not considered here, the *bile canaliculi network*, which transports bile fluid containing digestive enzymes [57, 58] (for recent digital reconstructions see [13, 14, 22]). Like the sinusoidal network, the bile canaliculi network spans across the entire liver lobule and contacts every single hepatocyte. Yet, bile fluid is toxic and must not enter the blood-stream, hence the two networks should nowhere get too close to each other. In two space dimensions, these competing design requirements could only be fulfilled by 'lines' of alternating network type, connecting source and sink. Such an architecture would only possess low resilience to damage. In three space dimensions, however, parallel network layers of alternating network type are possible. Such a design confers high transport capacity and resilience to damage. Indeed, signatures of such layered order were recently identified in liver tissue [14]. It must be stressed, however, that the geometry of sinusoid and bile canaliculi networks is much more disordered than a regular design of alternating network layers, as expected from self-organized networks that form in a tissue, where cells such as hepatocytes continuously divide. Such disordered networks are characterized by an inhomogeneous distribution of currents, where, at least in our minimal transport model, a small number of edges carry substantially higher currents than the majority of edges. Low-current edges, even if they are dispensable for the permeability of the network in the absence of perturbations and contribute only partially to network resilience, play an important biological role nonetheless, e.g. for the uptake of metabolites from the blood-stream, which relies on diffusion through the fenestrated surface of sinusoids. We expect that also the low-current edges contribute to the supply of hepatocytes, provided these are connected to high-current edges by short distances. Future work will address the self-organization of pairs of space-filling, mutually repulsive, intertwined networks [59], and study their transport and resilience properties, inspired by the bile and sinusoidal network in liver tissue.

## Methods

### Data acquisition

As described previously [13, 14, 35], fixed tissue samples of murine liver were optically cleared and treated with fluorescent antibodies for fibronectin and laminin, thus staining the extracellular matrix surrounding the sinusoids. Subsequently, samples were imaged at high-resolution using a multiphoton laser-scanning microscopy. Three-dimensional image data was segmented and network skeletons computed using MotionTracking image analysis software [35]. The data sets analyzed in this study correspond to the same used in [14].

### Hard sphere packing model

We used Monte-Carlo simulations to compute the radial distribution function $g(r) = g(r; R_0, \eta)$ of a packing of equally sized hard spheres with radius $R_0$ and volume fraction $\eta$ ($\eta$ = 0.15, 0.20, 0.30, 0.35). In these simulations, $n = 10^4$ spheres were initially positioned at regular grid positions, then 2000 Monte-Carlo cycles were performed, where each cycle consisted of testing a Monte-Carlo move with Gaussian displacement (standard deviation $\sigma = R_0/3$) for each of the spheres (acceptance rate about 75%).

For efficient computation, we exploited the fact that the radial distribution functions scales as $g(\lambda r; \lambda R_0, \eta) = g(r; R_0, \eta)$ if the radius $R_0$ of the hard spheres is changed to a new value $\lambda R_0$. Spline interpolation was used to interpolate $g(r; R_0, \eta)$ for intermediate values of $\eta$. A fit of the radial distribution function $g(r; R_0, \eta)$ for the minimal hard sphere model and the radial distribution function of the sinusoidal network resulted in $R_0 = 9.0405~\mu$m and $\eta = 0.2135~\mu$m$^{-3}$.

In a final step, a subset of sphere positions was selected to match the node density of the sinusoidal network. For Fig 2, a total of 1643 nodes were selected in a region of dimensions 420 $\mu$m $\times$ 450 $\mu$m $\times$ 90 $\mu$m, corresponding to a node density of $0.9659 \cdot 10^5$ mm$^{-3}$, which equals the node density of the entire sinusoidal network data set, including the void spaces occupied by the portal and central veins. For Fig 3, a total of 1964 nodes were selected in a region of dimensions 250 $\mu$m $\times$ 250 $\mu$m $\times$ 250 $\mu$m, corresponding to a node density of $1.2571 \cdot 10^5$ mm$^{-3}$, which equals the node density of the sinusoidal network excluding the void spaces occupied by the portal and central veins. Note that the random selection of a subset of hard sphere centers does not change the normalized radial distribution function $g(r)$.

For the set of simulated node positions $P_{sim}$, we computed the Delaunay graph $D(P_{sim})$ and the minimum spanning tree $M(P_{sim})$. As a technical note, the computation of the Delaunay graph $D(P_{sim})$ can generate artifacts at the boundaries of the simulation domain with unusually long edges. To avoid this, we first mirrored all nodes in the set $P_{sim}$ at the 6 planes defined by the boundaries, thereby obtaining an enlarged set of nodes and then computed the Delaunay graph for this enlarged set. Finally, we retained only edges that connected original nodes, while discarding those that connect to one or two mirrored nodes.

### Multi-objective optimization

We used simulated annealing to determine an optimal set of edges $E_\alpha^*$ that minimizes the multi-objective cost function $C_\alpha$ defined in Eq (4) for a given value of $\alpha$ with $0 \leq \alpha \leq 1$.

The optimization was initialized by a set of edges $E_{sim} = E_0$ where $E_0 = M(P_{sim}) \cup E_{rand}$ comprises the minimum spanning tree $M(P_{sim})$ and a random selection of edges $E_{rand} \subseteq D(P_{sim}) \setminus M(P_{sim})$ chosen from the set difference of the Delaunay graph and the minimum spanning tree of the set of node positions $P_{sim}$. For the simulations shown in Fig 2, the number of edges

in $E_{\text{rand}}$ was chosen to match the number $|E \setminus M(P)|$ of edges of the sinusoid data set that do not belong to the minimum spanning tree. For the simulations shown in Fig 3, this number $|E_{\text{rand}}|$ was chosen such that volume density of edges agree for $E_0$ and $E$.

In each step of the optimization procedure, we randomly selected a Monte-Carlo move that swaps a random pair of edges $e_1 \in E_{\text{sim}}$ and $e_2 \in D(P_{\text{sim}}) \setminus E_{\text{sim}}$. This Monte-Carlo move is accepted with probability $\exp(-\Delta C / T)$ if $\Delta C > 0$, and with probability 1 if $\Delta C < 0$, resulting in an update of the set of edges $E_{\text{sim}}$. Here, $\Delta C$ is the change in the normalized cost function that would result from this move and $T$ an effective temperature parameter.

The temperature parameter $T$ was initially set to 1 and kept constant for an initial melting phase of $10^4$ steps. Subsequently, $T$ was reduced by a factor of 0.998 after every 1000 steps, until a final temperature of $10^{-10}$ was reached. As validation tests, we confirmed that (i) the cost of the final network obtained at the end the annealing procedure equals the cost of the minimal-cost network encountered during the entire MC-optimization, (ii) multiple runs gave almost identical values of the minimal cost, and (iii) the computed Pareto front is continuous and concave, as predicted by theory. To account for the difference in range of the individual cost functions, $C_d$ and $C_e$, we used a normalized total cost function with linear rescaling $\bar{C}_{\bar{\alpha}}$ in the simulations, defined as

$$\bar{C}_{\bar{\alpha}} = \bar{\alpha}\bar{C}_d + (1 - \bar{\alpha})\bar{C}_e. \tag{10}$$

where the normalized individual cost functions read $\bar{C}_d = (C_d - \mathcal{U}_d)/(\mathcal{N}_d - \mathcal{U}_d)$ and $\bar{C}_e = (C_e - \mathcal{U}_e)/(\mathcal{N}_e - \mathcal{U}_e)$. The utopia and nadir points, $\mathcal{U}_d, \mathcal{U}_e, \mathcal{N}_d, \mathcal{N}_e$ were determined in preliminary simulations. The normalized cost functions satisfy $0 \leq \bar{C}_d[E^*_{\text{sim}}(\alpha)] \leq 1$ and $0 \leq \bar{C}_e[E^*_{\text{sim}}(\alpha)] \leq 1$ for the set of edges $E^*_{\text{sim}}(\alpha)$ that minimize $\bar{C}_\alpha$. We have the linear relationship $[\alpha(\mathcal{N}_d - \mathcal{U}_d) + (1 - \alpha)(\mathcal{N}_e - \mathcal{U}_e)]\bar{C}_{\bar{\alpha}} + \alpha\mathcal{U}_d + (1 - \alpha)\mathcal{U}_e = C_\alpha$, where $\alpha = (\mathcal{N}_e - \mathcal{U}_e)\bar{\alpha}/[(1 - \bar{\alpha})(\mathcal{N}_d - \mathcal{U}_d) + \bar{\alpha}(\mathcal{N}_e - \mathcal{U}_e)]$. In Fig 2E, we used the range $10^{-4} \leq \bar{\alpha} \leq 1 - 10^{-4}$. The value $\bar{\alpha} = 0.5$ corresponds to the value $\alpha^* = (\mathcal{N}_e - \mathcal{U}_e)/(\mathcal{N}_d - \mathcal{U}_d + \mathcal{N}_e - \mathcal{U}_e)$ for the cost function defined in Eq (4), and was used in all figures, unless stated otherwise.

## Supporting information

**S1 Movie. Three-dimensional visualization of sinusoidal network.** The supporting video shows a three-dimensional representation of the sinusoidal network shown in Fig 1C. (MP4)

**S1 Data. Sinusoidal networks data file.** The supporting data file is a zip-archive containing for three sinusoidal networks ASCII files with node positions of branch points, adjacency matrices and edge weights, respectively, as well as a file 'readme.txt' with information on the data format. The sinusoidal networks correspond to experimental data sets of adult murine liver tissue originally described in [14]. All three network sekeletons were processed as described in the main text. (ZIP)

**S1 Appendix. Simulation and data analysis methods.** The supporting appendix contains details on simulation and data analysis methods, as well as a minimal model of an adaptive network as discussed in the main text. (PDF)

## Acknowledgments

We thank the Centre for Information Services and High Performance Computing (ZIH) of the TU Dresden for the generous provision of computing power. We thank Lutz Brusch, Szabolcs Horvát, Felix Kramer, Steffen Lange, Kirstin Meyer, Carl Modes, Malte Schröder, Fabian Segovia-Miranda, Marc Timme, as well as all members of the Biological Algorithms group for stimulating discussions.

## Author Contributions

**Conceptualization:** Jens Karschau, Jonathan Wise, Yannis Kalaidzidis.

**Data curation:** Jens Karschau, Jonathan Wise, Hernán Morales-Navarrete.

**Funding acquisition:** Marino Zerial, Benjamin M. Friedrich.

**Investigation:** Jens Karschau, André Scholich, Hernán Morales-Navarrete.

**Methodology:** Benjamin M. Friedrich.

**Project administration:** Benjamin M. Friedrich.

**Resources:** Hernán Morales-Navarrete, Yannis Kalaidzidis.

**Software:** Jens Karschau, André Scholich, Jonathan Wise, Yannis Kalaidzidis, Benjamin M. Friedrich.

**Supervision:** Benjamin M. Friedrich.

**Visualization:** Jens Karschau, Hernán Morales-Navarrete, Benjamin M. Friedrich.

**Writing – original draft:** Jens Karschau, Benjamin M. Friedrich.

**Writing – review & editing:** Jens Karschau, Hernán Morales-Navarrete, Yannis Kalaidzidis, Marino Zerial, Benjamin M. Friedrich.

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
