## [Decision Letter · Decision Letter 0]

6 Feb 2020

Dear Dr. Friedrich,

Thank you very much for submitting your manuscript "Resilience of three-dimensional sinusoidal networks in liver tissue" for consideration at PLOS Computational Biology.

As with all papers reviewed by the journal, your manuscript was reviewed by members of the editorial board and by several independent reviewers. In light of the reviews (below this email), we would like to invite the resubmission of a revised version that takes into account the reviewers' comments.

We cannot make any decision about publication until we have seen the revised manuscript and your response to the reviewers' comments. Your revised manuscript is also likely to be sent to reviewers for further evaluation.

Sincerely,

Daniel A Beard

Deputy Editor

PLOS Computational Biology

Reviewer's Responses to Questions

**Comments to the Authors:**

Reviewer #1: In their article ``Resilience of three-dimensional sinusoidal networks in liver tissue,'' Karschau and co-workers investigate the structure and topology of sinusoidal networks in the liver. Based on data from their Ref. 13, the authors construct an algorithm for in-silico generation of arbitrarily large transport networks with the same

statistical properties as their real data. They consider both geometry (position of branch points as quantified through a radial distribution function) and topology (node degree and edge length distribution). In a second step, the authors include nematic alignment into their generation algorithm.

Based on their statistical model, the authors proceed to characterize resilience of their statistical and real networks to removal of edges. They find that while the networks are generically resilient to removal

of random edges, the loss of certain high-flow edges can lead to larger damage, as quantified by a loss in overall hydraulic permeability. Furthermore, the authors find that weak nematic alignment of the edges

with overall flow direction can increase permeability while retaining robustness.

The paper is generally well-written and interesting. The network generation method appears to be sound in terms of results.

However, I have a few concerns and remarks regarding details.

(*) While the authors employ multi-objective optimization techniques to find a Pareto front of trade-offs between their objectives, this Pareto front is then never actually used. In fact, after re-scaling the objectives appropriately, the authors end up simply averaging the individual objectives.

Given that the goal was to reproduce the statistics of their real data, scaling and averaging seems to be the most straightforward and direct fitting method.

I suspect that the normalization involving utopia and nadir points is of relevance here, but why these are chosen should be made more clear. So, while the Pareto terminology is interesting, it seems to me that

it might not actually needed to obtain the results in the paper. This is consistent with the authors' observation that their algorithm is robust with respect to choice of alpha.

(*) The authors conclude that ``weakly nematic networks optimize performance without compromising resilience.'' While the authors' results support this statement, here, Pareto methods would appear to be actually the appropriate tool for quantification. In an objective space comprising performance and permeability-at-risk, networks with weak nematic alignment should be on the Pareto front,

and those without should not be Pareto efficient.

(*) The authors mention that flows in their simulated networks are very inhomogeneous, contrary to intuition.

Is it possible that this inhomogeneity is related to their choice of pressure boundary conditions?

More specifically, what motivates the choice of fixing pressures instead of flows at the boundaries, and would fixing boundary flows lead to more homogeneous bulk flows?

(*) While available in Ref. 13, it would be useful to briefly mention again the size of the original data set in terms of both number of samples and size of the networks. How much larger than the real networks are

the authors' artificial ones? How large is the statistical advantage of using artificial networks over

the real data?

(*) The plots in Fig. 1 DEF suggest that the real data does not provide particularly smooth distributions (in particular for the edge lengths). What precautions have been taken to avoid overfitting of statistical

fluctuations in the data? How large is the error in the PDFs obtained from the real data?

Minor typos:

author summary:

vene, leafs -> vein, leaves

p. 10: extend -> extent

Reviewer #2: uploaded as an attachment

**Have all data underlying the figures and results presented in the manuscript been provided?**

Reviewer #1: Yes

Reviewer #2: None

PLOS authors have the option to publish the peer review history of their article (what does this mean?). If published, this will include your full peer review and any attached files.

Reviewer #1: No

Reviewer #2: Yes: Andrew D. Marquis
---

## [Decision Letter · Decision Letter 1]

15 May 2020

Dear PD Dr. Friedrich,

Thank you very much for submitting your manuscript "Resilience of three-dimensional sinusoidal networks in liver tissue" for consideration at PLOS Computational Biology. As with all papers reviewed by the journal, your manuscript was reviewed by members of the editorial board and by several independent reviewers. The reviewers appreciated the attention to an important topic. Based on the reviews, we are likely to accept this manuscript for publication, providing that you modify the manuscript according to the review recommendations.

Sincerely,

Daniel A Beard

Deputy Editor

PLOS Computational Biology

[LINK]

Reviewer's Responses to Questions

**Comments to the Authors:**

Reviewer #1: I am satisfied with the changes made by the authors and recommend publication.

If the authors would like to add the new supplemental figures they mentioned at a later stage that could be good, but I feel that the paper is sufficiently well supported at this time.

Reviewer #2:

**General**

              Overall, the authors have done an excellent job with this resubmission. I am particularly impressed with the addition of the minimal model of an adaptive network, and the additional details on Pareto optimality brought up by the other reviewer. I only have one major critique and have identified a few typos and clarity suggestions. Treat my typo and clarity suggestions as optional. This is an interesting and well-done study and I am looking forward to seeing it published.

**Critiques**

In the “Minimal model for adaptive edge weights” section, you describe a model that regulates edge permeability in a way that is dependent upon edge current going below a set point (I*). How was this set point determined? Are the results sensitive to the choice of set point, I*?

**Typos and clarity suggestions**

In the 7^th^ paragraph of the introduction, you say “Permeabilities allow to efficiently … ”, and I think meant to say “Permeabilities allow us to efficiently …”In the 4^th^ paragraph of the “A network generation algorithm for spatial networks” section, you say “which is in particular robust …”, and I think meant to say “which is particularly robust …”In the first paragraph of the “Discussion and Outlook” section, you say “We drew advantage …” and I think meant to say “We took advantage …”In the 6^th^ paragraph of the “Discussion and Outlook” section, you say “may facilitate a more homogenous distribution instead.” – I think you can delete the word “instead” from this sentenceIn the “Permeability-at-risk if low-current edges removed” section, I think the section could be re-named “Permeability-at-risk if low-current edges are removed”In the “Permeability-at-risk if low-current edges removed” section, you say “However, if together …” and I think you meant to say “However, if performed together”In equation S3, it appears that kappa_0 cancels out. Is this case?Your reasoning behind the use of kappa/kappa_0 as a criterion to constrain the change in permeability by a factor of 4 is sound, but perhaps you can express equations S3 as a piecewise function where dkappa_j/dt = -rho kappa (I* - |I_j|) if 0.25 < kappa/kappa_0 < 4, and dkappa_j/dt = 0 otherwise**********

**Have all data underlying the figures and results presented in the manuscript been provided?**

Reviewer #1: Yes

Reviewer #2: Yes

PLOS authors have the option to publish the peer review history of their article (what does this mean?). If published, this will include your full peer review and any attached files.

Reviewer #1: No

Reviewer #2: Yes: Andrew D. Marquis
---

## [Editor Report · Decision Letter 2]

19 May 2020

Dear PD Dr. Friedrich,

We are pleased to inform you that your manuscript 'Resilience of three-dimensional sinusoidal networks in liver tissue' has been provisionally accepted for publication in PLOS Computational Biology.

Best regards,

Daniel A Beard

Deputy Editor

PLOS Computational Biology

Daniel Beard

Deputy Editor

PLOS Computational Biology

---

## [Editor Report · Acceptance letter]

19 Jun 2020

PCOMPBIOL-D-19-02210R2 

Resilience of three-dimensional sinusoidal networks in liver tissue

Dear Dr Friedrich,

I am pleased to inform you that your manuscript has been formally accepted for publication in PLOS Computational Biology. Your manuscript is now with our production department and you will be notified of the publication date in due course.

With kind regards,

Laura Mallard
